Metabolomic and biochemical characterization of a new model of the transition of acute kidney injury to chronic kidney disease induced by folic acid

Perales-Quintana Marlene Marisol 1
http://orcid.org/0000-0002-1028-9122 Saucedo Alma L. 2
Lucio-Gutiérrez Juan Ricardo 2
Waksman Noemí 2
Alarcon-Galvan Gabriela 3
Govea-Torres Gustavo 4
Sanchez-Martinez Concepcion 5
Pérez-Rodríguez Edelmiro 6
http://orcid.org/0000-0001-7925-6524 Guzman-de la Garza Francisco J. 1
Cordero-Pérez Paula 4 paucordero@yahoo.com.mx
1 Physiology Department, Universidad Autónoma de Nuevo León , Monterrey, Nuevo León , Mexico
2 Analytic Chemistry Department, Universidad Autónoma de Nuevo León , Monterrey, Nuevo León , Mexico
3 Basic Science Department, School of Medicine, Universidad de Monterrey , Monterrey, Nuevo León , Mexico
4 Liver Unit, Department of Internal Medicine, “Dr. José E. González” University Hospital, Universidad Autónoma de Nuevo León , Monterrey, Nuevo León , Mexico
5 Nephrology Department, “Dr. José E. González” University Hospital, Universidad Autónoma de Nuevo León , Monterrey, Nuevo León , Mexico
6 Transplant Service, “Dr. José E. González” University Hospital, Universidad Autónoma de Nuevo León , Monterrey, Nuevo León , Mexico
Kennedy David
Electronic publication date: 2019 Jun 21
Publication date: 2019
Volume: 7
Electronic Location ID: e7113
Received 2019 Jan 30; Accepted 2019 May 10
Copyright: © 2019 Perales-Quintana et al.
Copyright year: 2019
Copyright holder: Perales-Quintana et al.
License: This is an open access article distributed under the terms of the Creative Commons Attribution License, which permits unrestricted use, distribution, reproduction and adaptation in any medium and for any purpose provided that it is properly attributed. For attribution, the original author(s), title, publication source (PeerJ) and either DOI or URL of the article must be cited.
License URL: https://creativecommons.org/licenses/by/4.0/

Keywords: Folic acid, Chronic kidney disease, PCA, NMR metabolomics, Acute kidney disease

Funding: National Council on Science and Technology—Conacyt (project 2017-01-5652) This work was supported by National Council on Science and Technology–Conacyt (project 2017-01-5652). The funders had no role in study design, data collection and analysis, decision to publish, or preparation of the manuscript.

==============================
Background

Renal diseases represent a major public health problem. The demonstration that maladaptive repair of acute kidney injury (AKI) can lead to the development of chronic kidney disease (CKD) and end-stage renal disease has generated interest in studying the pathophysiological pathways involved. Animal models of AKI–CKD transition represent important tools to study this pathology. We hypothesized that the administration of multiple doses of folic acid (FA) would lead to a progressive loss of renal function that could be characterized through biochemical parameters, histological classification and nuclear magnetic resonance (NMR) profiling.

Methods

Wistar rats were divided into groups: the control group received a daily intraperitoneal (I.P.) injection of double-distilled water, the experimental group received a daily I.P. injection of FA (250 mg kg body weight−1). Disease was classified according to blood urea nitrogen level: mild (40–80 mg dL−1), moderate (100–200 mg dL−1) and severe (>200 mg dL−1). We analyzed through biochemical parameters, histological classification and NMR profiling.

Results

Biochemical markers, pro-inflammatory cytokines and kidney injury biomarkers differed significantly (P < 0.05) between control and experimental groups. Histology revealed that as damage progressed, the degree of tubular injury increased, and the inflammatory infiltrate was more evident. NMR metabolomics and chemometrics revealed differences in urinary metabolites associated with CKD progression. The main physiological pathways affected were those involved in energy production and amino-acid metabolism, together with organic osmolytes. These data suggest that multiple administrations of FA induce a reproducible model of the induction of CKD. This model could help to evaluate new strategies for nephroprotection that could be applied in the clinic.

Introduction

Interest in kidney health has recently increased mainly because of the alarming statistics. It has been estimated that 5–10 million people die annually from either chronic kidney disease (CKD) or acute kidney injury (AKI) (Luyckx, Tonelli & Stanifer, 2018). Previous studies have estimated a worldwide prevalence of 8–16% for CKD (Jha et al., 2013) and 1–25% (Lameire et al., 2013) for AKI. Although these two diseases were originally considered to be totally unrelated syndromes, the evidence suggests that maladaptive repair from AKI can lead to the development of CKD, and that either of these two syndromes can lead to the development of end-stage renal disease (Chawla et al., 2014). At present, there is no specific treatment to reverse or stop kidney disease. The currently used treatments attempt to identify the syndromes at an early stage and arrest or delay their natural history of progression before the need for dialysis or transplantation (Eknoyan et al., 2004).

However, we cannot ignore that the cost of treating kidney diseases and their associated complications is a challenge for health services around the world; for example, the annual costs of CKD treatment range between US$35,000 and $100,000 per patient (Levin et al., 2017).

In clinical practice, urea and creatinine levels have been traditionally used as markers of renal function for early identification of kidney disease. However, it has been reported that these tests have limitations and may under-estimate the disease stage. To overcome this restriction, exploration using “omics” sciences (such as proteomics and metabolomics) has allowed the identification of new non-invasive biomarkers of renal function (Saucedo-Yanez et al., 2018).

Studies have suggested that a persistent inflammatory response mediated by pro-inflammatory cytokines such as interleukin 1 beta (IL-1β), interleukin 6 (IL-6) and tumor necrosis factor alpha (TNF-α) contributes to the perpetuation of kidney damage (Furuichi, Kaneko & Wada, 2009). Other proteins including kidney injury molecule-1 (KIM-1), neutrophil gelatinase associated with lipocalin (NGAL) and cystatin C (Cys-C) also have a major impact on the development of kidney disease. KIM-1, NGAL and Cys-C were originally proposed as markers of AKI, but recently have been used to describe the evolution of CKD (Gil et al., 2016; Lobato et al., 2017). In addition, the omics sciences allow us to evaluate globally the different molecules of a family, rather than a single protein or metabolite, in such a way that the physiological state of an organism can be described by the presence or absence of these molecules (Amaro et al., 2016). In particular, developments in metabolomics allow the identification of metabolites in a living organism, and this chemical fingerprint can help to detect imbalances in metabolic pathways as a result of physiological or pathological changes. The use of technologies, such as mass spectrometry and proton nuclear magnetic resonance (NMR) spectroscopy, in sum with bioinformatic data analyses, are key for the identification of metabolites for a prospective use as biomarker of some pathology. Early metabolomic studies, conducted either on experimental models or on patients, had retrieved a set of different metabolites that could be used for the early diagnosis or prognosis of different renal diseases (Hocher & Adamski, 2017; Kimura et al., 2016; Perales-Quintana et al., 2017; Zhang et al., 2016; Zhao, 2013). For examples, recent analysis in blood samples of CKD has identified metabolic changes in levels of glucose, citrate, lactate, valine, alanine, glutamate, glycine, betaine, myo-inositol, taurine, glycerylphosphorylcholine and trimethylamine-N-oxide (TMAO) (Lee et al., 2016; Qi et al., 2012; Rhee et al., 2013). In another study, urine analysis identified 5-oxoproline, glutamate, guanidoacetate, α-phenylacetylglutamine, taurine, citrate and TMAO as a panel of metabolites that might assist in the identification and monitoring of patients with CKD (Posada-Ayala et al., 2014). In this regard, the search for markers of AKI have been more challenging due to its multifactorial origin. Blood metabolic profile revealed that acylcarnitines, methionine, homocysteine, pyroglutamate, asymmetric dimethylarginine, and phenylalanine, arginine and several lysophosphatidyl cholines were disturbed in patients with AKI, when compared to healthy subjects (Sun et al., 2012). It could be inferred through urine analysis that 2-hydroxybutyric acid, pantothenic acid, hippuric acid, N-acetylneuraminic acid, phosphoethanolamine, and serine may be related to pathophysiological changes associated with AKI (Martin-Lorenzo et al., 2017).

Animal models are key to allowing further examination of renal function, to study disease pathogenesis in an accelerated time frame and to assist in the search for therapeutic approaches. Different models of AKI to CKD transition have been proposed, including ischaemia–reperfusion injury (IRI) (Le Clef et al., 2016) or repeated administration of drugs such as cisplatin (Sharp et al., 2018) or aristolochic acid (Debelle et al., 2002). In particular, the injection of folic acid (FA) at high doses is an established model for induction of AKI, because the low solubility of FA results in formation of crystals in the renal lumen (Schmidt et al., 1973). This has been suggested to lead to an alteration in cellular architecture and generate oxidative stress and fibrosis that could progress to CKD (Stallons, Whitaker & Schnellmann, 2014). Fu et al. (2018) hypothesized that the repeated injury by multiple administrations of FA could be sufficient to cause the transition from AKI to CKD.

To establish whether the pharmacological AKI induced by repetitive FA dosing progresses to CKD, the effect of FA on distinctive parameters of renal function must be explored. The objective of this study was to characterize in rats the progression of CKD induced by multiple administrations of FA leading to mild, moderate and severe uraemia, in a way that provided a wider view of the modifications in biomarkers including pro-inflammatory cytokines, urine metabolomic profile and histological changes during the transition from physiological to pathophysiological states.

Materials and Methods

Ethical approval

All animal experiments were performed in accordance with the principles and regulations of the Mexican Official Norm NOM-062-ZOO-1999 and were approved by the Animal Care and Use Committee of our institution (HI17-00004).

Animals and experimental design

Male Wistar rats (200–300 g) were obtained from a colony raised in the Círculo A.D.N. S.A. de C.V., Mexico City, Mexico and were housed in clean polypropylene cages under a controlled 12-h light–dark cycle at a stable room temperature (24 ± 3 °C) and had access to commercial rat pellets and water ad libitum.

Animals were divided into two groups, the control group and the FA injection group (FAIG). The FAIG was further subdivided into three groups based on the severity of disease induced: mild (FAIG-Mi), moderate (FAIG-Mo) and severe (FAIG-S), with each group containing six animals, giving a total of 24 experimental animals.

Before the study, all rats were allowed a 7-day adaptation phase. The study comprised an induction phase (day 0–9), during which the animals in the FAIG were administered an intraperitoneal (I.P.) injection of FA (250 mg kg body weight−1; Sigma, Poole, Dorset, UK) every third day, followed by a maintenance phase (day 10 to the end of the experiment) in which the animals in the FAIG received the same I.P. dose of FA daily. The rats in the control group were given the same I.P. volume of double-distilled water at the corresponding frequency in each phase.

Once a week during the maintenance phase, rats were placed in metabolic cages (one rat per cage) for 24-h urine collection; all samples were frozen at −80 °C until analysis. Blood samples from the lateral tail vein of the rat were obtained to assess renal function by measuring the levels of creatinine and blood urea nitrogen (BUN) on the same day. The degree of kidney damage in the FAIGs was classified based on the BUN values as established by Ormrod & Miller (1980): mild (40–80 mg dL−1), moderate (100–200 mg dL−1) and severe (>200 mg dL−1). The animals were divided into subgroups and anaesthetized with an I.P. injection of a mixture of ketamine (100 mg kg−1; Reg. SAGARPA Q7833-028, Anesket, PiSA Agropecuaria, S.A. De C.V., Atitalaquía, Mexico) and xylazine (10 mg kg−1; Reg. SAGARPA Q-0088-122, Sedaject, Vedilab S.A. De C.V., Mexico City, Mexico). Depth of anaesthesia was assessed by testing the pedal reflex (firm toe pinch), then the rats were killed by cardiac puncture and exsanguination. Blood samples were taken and centrifuged at 3,500 rpm for 15 min (SIGMA 2-5 Centrifuge, Osterode am Harz, Germany). The serum was separated and then stored at −80 °C until use. Both kidneys were removed and placed in 10% formalin for subsequent histological evaluation.

Biochemical markers, pro-inflammatory cytokines and kidney injury biomarkers

The concentrations of creatinine and urea in serum were determined using a modified Jaffe colorimetric method and the urease enzymatic method, respectively, by means of commercial kits and spectrophotometry with an automatic analyser ILAB-Aries (ILab-300 Plus; Instrumentation Laboratory, Bedford, MA, USA).

The serum levels of pro-inflammatory cytokines IL-1β, IL-6 and TNF-α were determined using a sandwich enzyme-linked immunoassay protocol development kit (Peprotech, Mexico City, Mexico). The optical density was measured at 405/620 nm using a microplate-reading spectrophotometer (Mutilskan FC, Thermo Fischer Scientific Oy, Vantaa, Finland). The results are expressed as ng/mL.

For quantitative detection of rat serum NGAL and KIM-1, Abcam one-step ELISA kits were used (Lipocalin-2/NGAL Rat ELISA Kit, Abcam 119602, Abcam, Cambridge, UK; KIM-1/TIM-1 Rat ELISA Kit Abcam 119597, Cambridge, MA, USA, respectively). Cys-C was measured using ELISA kits from R&D Systems (Quantikine ELISA; Mouse/Rat Cystatin C Immunoassay, R&D Systems, Minneapolis, MN, USA).

Renal histopathology

Kidneys from all rats of each group were fixed in 10% buffered formaldehyde solution (pH 7.4). After fixation, the tissue was embedded in paraffin and cut into 4-μm sections, which were deparaffinized and stained with haematoxylin and eosin (H&E). These sections were subsequently evaluated under an optical microscope for indicators of cell damage, such as: the increase in Bowman’s space area, degenerative tubular changes (edema and cytoplasmic vacuolization), tubular dilation, tubular necrosis, vascular congestion, intratubular proteins and neutrophil casts, leukocyte interstitial infiltration, leukocyte intratubular infiltration, interstitial fibrosis and tubular atrophy. Damage was evaluated using a scale ranging from zero to five: not present (grade 0), 1–20% injuries (grade 1), 21–40% injuries (grade 2), 41–60% (grade 3), 61–80% (grade 4) and 81–100% (grade 5). Lastly, the total histopathologic score was calculated, as the sum of all grades of the different injuries.

Preparation of samples and acquisition of 1H-NMR data

To proceed with NMR experiments, urine samples were thawed at room temperature and vortexed to homogenize. A volume of 500 μL of urine was mixed with 60 μL D2O containing 3-(trimethylsilyl) propionic-2,2,3,3-d4 acid sodium salt (TSP) (0.75% v/v; Sigma, Poole, Dorset, UK) and 40 μL phosphate buffer (1.5 M, pH 7.0; Na2HPO4/NaH2PO4; Sigma, Poole, Dorset, UK) to minimize variations in the pH of samples. Then, the samples were vortexed and centrifuged at 13,000 g for 5 min. Finally, 550 μL of supernatant was transferred to a five-mm NMR tube.

Nuclear magnetic resonance spectra of samples were collected at 298 K on an Bruker Avance III HD 400 MHz (Bruker Biospin, Billerica, MA, USA) spectrometer equipped with a BBO SmartProbe with z-gradients. The 1D nuclear Overhauser enhancement spectroscopy experiment with pre-saturation for water signal (1D-NOESYpresat) was used with a fixed relaxation delay. For each sample, 128 transients were collected into 32,768 data points with a spectral width of 21.04 ppm (8,417.509 Hz) and an acquisition time of 1.95 s. All spectra were phased, and baseline corrected and referenced to the TSP peak.

Resonance assignments were performed based on the literature in combination with data from the Human Metabolomic Database (http://www.hmdb.ca/) and the Biological Magnetic Resonance Bank (http://www.bmrb.wisc.edu/) and confirmed with two-dimensional NMR spectra of representative samples. Correlation spectroscopy data were acquired using the cosyprqf pulse sequence with 16 scans per 128 increments, a 10.3 ppm spectral width, and the transmitter frequency offset at 4.7 ppm. Total correlation spectroscopy data were acquired with the mlevphpr.2 pulse sequence with 16 scans per 256 increments, a 10.0 ppm spectral width, the transmitter frequency offset at 4.7 ppm, and a mixing time of 200 ms to optimize the magnetization transfer to the whole spin system of each metabolite. Heteronuclear single-quantum correlation data were acquired with the hsqcetgpsisp2.2 pulse sequence using 16 scans per 256 transients, spectral widths of 16 and 180 ppm and offsets of 4.7 and 75 ppm for the 1H and 13C dimensions, respectively. This sequence was optimized for direct coupling constants of 145 Hz.

Statistical analysis

Data for body weight, urine output, creatinine and BUN levels are expressed as the mean ± SD and were analyzed by one-way analysis of variance followed by Tukey’s test for multiple comparisons; significance was set at P < 0.05. Analysis was performed using GraphPad Prism software (v. 6.0; GraphPad, San Diego, CA, USA).

Multivariate pattern recognition analysis

All spectra were exported from the NMR instrument as JCAMP-DX files and then imported to the MATLAB (MathWorks, Natick, MA, USA) computing environment for data handling. The PLS Toolbox software (Eigenvector Research) was used for data pretreatment and mathematical model construction. The regions containing the residual peak from the suppressed water resonance (4.68–5.05 ppm) together with the initial and final regions of the spectra (0.0–0.5 and 9.5–21.0 ppm, respectively) were excluded because they were uninformative. The resulting data vectors were co-added, that is, adjacent variables in each spectrum were combined, using the mean value of two variables. The interval correlation shifting (icoshift) algorithm (Savorani, Tomasi & Engelsen, 2010) was used for spectral alignment of each experimental group (Control, FAIG-Mi, FAIG-Mo and FAIG-S). For this, the NMR spectra were split into 45 intervals, which were selected by visual inspection based on the presence of signals. The “max” and “b” options were selected to define the reference vector and to allow the algorithm to search for the best maximum shift correction in data, respectively. To avoid any influence on the results of multivariate analysis resulting from FA signals in urine spectra, the following regions were also removed: 1.95–2.19; 2.23–2.37; 4.25–4.35; 4.46–4.49; 6.74–6.79; 7.60–7.70; 8.05–8.12; and 8.68–8.72 ppm. Subsequently, the spectra were sample-wise normalized using the probabilistic quotient normalization (PQN) (Dieterle et al., 2006) and mean centered. PQN was preferred since it is a robust normalization method, similar to the multiplicative scatter correction but using the median as the target and a fitting of each row to the target Different metabolomics papers using urine from human (Kostidis et al., 2019; Pinto et al., 2015) or murine models (Pelantová et al., 2015; Torres Santiago et al., 2019), have reported the use of this normalization technique.

Data were then subjected to principal component analysis (PCA) to identify the natural grouping tendency of the samples and for assessment of the presence of outliers. Cross validation was used when the calibration model was developed. The optimum number of principal components (PCs) was determined by the minimum value of the predicted residual error sum of squares criterion. Statistics calculated for the calibration model included the root mean square error of cross validation (RMSECV). Score plots of the first three PCs were used to visualize the separation of the clusters, whereas the loading plots were used to identify the spectral variables that contributed to the separation between clusters. We selected loadings >0.05 for those contributing to the differentiation of clusters.

To compare different models, and analytical platforms, analysis of the metabolic pathways were performed with Metaboanalyst (https://www.metaboanalyst.ca/), using the metabolites that corresponded to the variables that contributed to the separation between each group.

Results

Biochemical markers, pro-inflammatory cytokines and kidney injury biomarkers

The serum BUN and creatinine of the control group were significantly lower than those of the FAIGs (P < 0.0001 each). Furthermore, the BUN and creatinine levels gradually increased as kidney injury worsened.

The serum levels of IL-1β and IL-6 showed similar trends: the control group differed significantly only from that in the FAIG-Mo and FAIG-S subgroups; moreover, these damage groups differed significantly from the FAIG-Mi group. In contrast, in TNF-α the levels differed significantly between the control group and the FAIGs, and there were significant differences between each stage of damage (Fig. 1).

Figure 1 Serum levels of biochemical markers, pro-inflammatory cytokines and kidney injury biomarkers.

(A) Serum level of BUN; (B) Serum level of creatinine; (C) Serum level of interleukin 1β; (D) Serum level of interleukin 6; (E) Serum level of tumor necrosis factor alpha; (F) Serum level of neutrophil gelatinase associated with lipocalin; (G) Serum level of kidney injury molecule-1; (H) Serum level of cystatin C. Values are expressed as means ± SD. N = 6 in each group. *P < 0.05 vs. C. †P < 0.05 vs. FAIG-Mi. #P < 0.05 vs. FAIG-Mo. FAIG-Mi: Mild damage, FAIG-Mo: Moderate damage, FAIG-S: Severe damage.

For the serum NGAL levels, two of the damage groups differed significantly from the control group: the levels were higher in the FAIG-Mi group but lower in the FAIG-S group. The serum NGAL levels decreased markedly with increasing damage. The serum KIM-1 level was significantly higher in all FAIGs than in the control group. Among the FAIG subgroups, the KIM-1 levels differed significantly between the FAIG-Mo and FAIG-S groups and between the FAIG-Mi and FAIG-S groups. Serum concentrations of Cys-C differed significantly between the control group and the FAIGs, but there was no significant difference between the FAIG subgroups (Fig. 1).

Renal histopathology

Representative renal histology of the different experimental groups is shown in Fig. 2. Histological score of kidney damage is shown in Table 1. Examination of H&E-stained tissue sections showed normal renal parenchyma, tubules and glomeruli in the control group (Fig. 2A). By contrast, the architecture of the kidneys from the FAIGs was affected, as it is summarized in Table 1. It is shown that total histological score increased in FAIGs, when compared to the control group. In the FAIG-Mi group, the interstitium was apparent because of the presence of an inflammatory mixed infiltrate, and the tubules were affected with degenerative changes and tubular dilatation (Fig. 2B). The FAIG-Mo group was characterized by decreased levels of preserved tissue, which was surrounded by an abundant inflammatory infiltrate. Degenerative tubular changes, tubular dilatation and tubular necrosis were present. In addition, intratubular casts become more apparent with an increase of interstitial fibrosis and tubular atrophy (Fig. 2C). The FAIG-S group showed isolated areas of preserved tissue and severe tubular dilatation and atrophy with the presence of cast-containing cellular debris and inflammatory infiltrate (Fig. 2D).

Figure 2 Representative light microphotographs of sections of renal tissue from each experimental group, stained with haematoxylin and eosin.

(A) Control group, (B) FAIG-Mi: Mild damage, (C) FAIG-Mo: Moderate damage, (D) FAIG-S: Severe damage.

Table 1 1H-NMR chemical shift assignment of differential metabolites observed in urine from control and FA-treated rats and their change trends (increased/decreased) compared with the control group.

Parameters	Groups	
Control	FAIG-Mi	FAIG-Mo	FAIG-S	
Bowman’s space enlargement	0.7	1.0	2.0	2.7	
Degenerative tubular changes	1.0	0.5	2.3	2.7	
Tubular dilatation	0.0	2.0	3.7	4.3	
Tubular necrosis	0.0	0.0	0.7	1.0	
Vascular congestion	1.0	0.5	0.7	1.3	
Intratubular proteins casts	1.0	1.0	1.7	1.3	
Intratubular neutrophil casts	0.0	0.5	2.0	1.7	
Leukocyte interstitial infiltration	0.7	2.5	3.7	4.0	
Leukocyte intratubular infiltration	0.0	1.0	2.7	2.3	
Interstitial fibrosis	0.0	0.5	2.0	3.0	
Tubular atrophy	0.0	2.0	3.0	3.3	
Total histopathologic score	4.3	11.5	24.3	27.6	
Note:

FAIG-Mi, Mild damage; FAIG-Mo, Moderate damage; FAIG-S, Severe damage.

Acquisition of 1H-NMR data and multivariate pattern recognition analysis

Representative 1H-NMR spectra of urine samples obtained from the control and FAIGs are shown in Fig. 3. Resonance assignments were made based on the literature in combination with the Human Metabolomic Database and the Biological Magnetic Resonance Bank and confirmed by two-dimensional NMR spectra. Table 2 lists the 23 metabolites identified, their chemical shift and comparison of their peak intensity relative to the control group. To facilitate the comparison of the different metabolites, subplots of the spectra for each experimental group were generated and fixed in an array of two rows by two columns. This arrangement made it possible to maintain the same spectral region at the same scale between the groups; an example is depicted in Fig. 4.

Figure 3 1H-NMR spectra of rat urine from the different experimental groups.

(A) 1H-NMR spectra of urine samples of each experimental group. (B) High field of the 1H-NMR spectra of urine samples of each experimental group. (C) Low field of the 1H-NMR spectra of urine samples of each experimental group. Identified metabolites: (1) valine, (2) lactate, (3) alanine, (4) acetate, (5) succinate, (6) oxoglutarate, (7) citrate, (8) dimethylamine, (9) trimethylamine, (10) creatinine, (11) creatine, (12) malonate, (13) taurine, (14) TMAO, (15) glycine, (16) phenylacetylglycine, (17) hippurate, (18) allantoin, (19) urea, (20) aconitate, (21) kynurenate, (22) n-1-methylnicotinamide (23) trigonelline. (*) Removed peaks correspond to the folic acid signals.

Table 2 Histopathological damage evaluation induced by repeated administration of folic acid.

Metabolite	HMDB ID	1H-NMR Chemical shift (ppm, multiplicity)	FAIG-Mi	FAIG-Mo	FAIG-S	
Energy metabolites	
 Acetate	HMDB0000042	1.92 (s)	↑	↑	↑	
 Aconitate	HMDB0000072	3.45 (d) 6.58 (s)	↑	↑	↑	
 Citrate	HMDB0000094	2.54 (d) 2.69 (d)	↓	↓	↓	
 Creatinine	HMDB0000562	3.03 (s) 4.05 (s)	↓	↓	↓	
 Lactate	HMDB0000190	1.33 (d)	↑	≈	≈	
 Oxoglutarate	HMDB0000208	2.44 (t) 3.01 (t)	≈	↓	↓	
 Succinate	HMDB0000254	2.41 (s)	↑	↓	↓	
Amino acid metabolism	
 Alanine	HMDB0000161	1.48 (d)	≈	≈	↓	
 Creatine	HMDB0000064	3.04 (s) 3.93 (s)	≈	≈	↑	
 Glycine	HMDB0000123	3.56 (s)	↑	↓	↓	
 Kynurenate	HMDB0000715	6.61 (s)	↓	↓	↓	
 Malonate	HMDB0000691	3.11 (s)	↓	↓	↓	
 Urea	HMDB0000294	5.80 (s)	≈	↓	↓	
 Valine	HMDB0000883	0.99 (dd)	↓	↓	↓	
Gut flora metabolite	
 Hippurate	HMDB0000714	3.97 (d) 7.55 (t) 7.64 (t) 7.83 (d)	↓	↓	↓	
Organic osmolytes	
 DMA	HMDB0000087	2.72 (s)	↓	↓	↓	
 TMA	HMDB0000906	2.89 (s)	↑	↑	↑	
 TMAO	HMDB0000925	3.27 (s)	↓	↓	↓	
 Taurine	HMDB0000251	3.27 (t) 3.43 (t)	↓	↓	↓	
Metabolism of cofactors and vitamin	
 NMN	HMDB0000699	9.27 (s) 8.99(d) 8.89 (d) 8.18 (t) 4.48 (s)	↓	↓	↓	
Other metabolites	
 Allantoin	HMDB0000462	5.39 (s)	↓	↓	↓	
 PAG	HMDB0000821	3.67 (s) 3.76 (d)	↓	↓	↓	
 Trigonelline	HMDB0000875	8.06 (tt) 8.83 (tt) 9.11 (s)	↓	↓	↓	
Note:

HMDB ID, Identification code in the Human Metabolomic Data Base; FAIG-Mi, Mild damage; FAIG-Mo, Moderate damage; FAIG-S, Severe damage; increase (↑) or decrease (↓) vs. control group; DMA, dimethylamine; TMA, trimethylamine; TMAO, Trimethylamine N oxide; NMN, N1-methylnicotinamide; PAG, Phenylacetylglycine.

Figure 4 Selected regions of the 1H-NMR spectra of urine from each experimental group.

The singlet of the creatinine signal (4.05 ppm) and the doublet of the signal of hippurate (3.97 ppm) in the (A) Control group, (B) FAIG-Mi, (C) FAIG-Mo, (D) FAIG-S. The acetate singlet (1.92 ppm) in the (E) Control group, (F) FAIG-Mi, (G) FAIG-Mo, (H) FAIG-S. The n-1-methylnicotinamide doublets (8.89 and 8.99 ppm) in the (I) Control group, (J) FAIG-Mi, (K) FAIG-Mo, (L) FAIG-S. Visual differences are clearly detectable.

To explain the differences in the urine metabolite profiles of each group, 1H-NMR spectra were subjected to PCA. It was necessary to pre-process the spectra data to remove unwanted variations resulting from instrumental instabilities, the effects of pH or from ionic strength because these factors could affect the interpretation of the multivariate analysis. Therefore, spectral data were reduced (co-added), uninformative areas were removed, and chemical-shift drifts were corrected by alignment (icoshift algorithm from Savorani et al., normalized (PQN) and mean centered). After each step, the spectra were evaluated by visual inspection and with a preliminary PCA; no obvious deterioration in the signals of sample spectra was observed. Under these experimental conditions, three PCs were chosen for building the PCA model because they produced the lowest RMSECV (0.0003251), with a cumulative explained variance of 64.32%.

The score plots of the PCA analysis of urine samples showed a clustering according to the experimental groups; this was expected and desirable behavior. Figure 5A shows that 3D scores plotted from PC1 (37.53%), PC2 (23.22%) and PC3 (14.16%) demonstrated well-differentiated regions for each group: FAIG-S was located by positive scores of PC1, the control group by positive scores of PC2 and the split between FAIG-Mi and FAIG-Mo was based on positive and negative PC3 scores, respectively. A narrow dispersion was observed for control samples, a moderate dispersion for FAIG-Mi and FAIG-Mo, and a broader dispersion for FAIG-S. Figures 5B and 5C show 2D score plots to allow better visualization. The loading plots illustrate the importance of each variable within original data. The contributions of each PC to the loading plots suggest the variables responsible for the observed agglomeration of the samples. The strongest contribution corresponded to signals for metabolites in the high field (1.0–4.2 ppm). The signals of the metabolites that were selected were those that had greater loads (>0.05) in the loading plot (Fig. 6). In the FAIG-Mi group, signals of acetate, glycine, lactate and succinate tended to increase when compared to the control group. Meanwhile, signals of citrate, creatine, creatinine and TMAO, tended to decrease. On the other hand, the most important contributions for the FAIG-Mo group correspond to a decrease in the signals of citrate, creatinine, Phenylacetylglycine (PAG) and TMAO. The increased signals of acetate and creatine, altogether with a decrease in the signals of allantoin, creatinine and taurine, contribute for the cluster in FAIG-S (Fig. 7).

Figure 5 Score plots from PCA applied to 1H-NMR spectra of rat urine samples from control rats and rats with AKI–CKD transition induced by folic acid.

(A) 3D plot of the three components. (B) Score plot of PC1 and PC2, (C) Score plot of PC1 and PC3. Control group: green circles, FAIG-mild group: yellow squares, FAIG-moderate group: orange triangles, FAIG-severe group: red diamonds.

Figure 6 Loading plots from PCA applied to 1H-NMR spectra of rat urine samples from control rats and rats with AKI–CKD transition induced by folic acid.

Loading plots from (A) PC1, first principal component; (B) PC2, second principal component and (C) PC3, third principal component.

Figure 7 Pathway analysis of identified metabolites in the different groups.

The upper panels (A–C) represent the relevant metabolic pathways on the basis of the urine metabolites of each group using the MetaboAnalyst. The lower tables (D–F) correspond to the metabolites whose NMR signal increased or decreased in comparison to control group. (A) and (D) FAIG-Mi: Mild damage; (B) and (E) FAIG-Mo: Moderate damage; (C) and (F) FAIG-S: Severe damage.

Metabolic pathway analysis using Metaboanalyst (https://www.metaboanalyst.ca/), led to the identification of several pathways significantly affected by the AKI-CKD transition induced by FA. The most relevant pathways in FAIG-Mi group were citrate cycle, pyruvate metabolism, glycolysis or gluconeogenesis and glycine serine and threonine metabolism. An overall evaluation showed that in the FAIG-Mo group the disturbs pathway correspond to both the citrate cycle, and the arginine and proline metabolism, while in the FAIG-S group taurine and arginine metabolism were the most relevant pathways (Fig. 7).

Discussion

In recent years, the study of the development of CKD from severe or repeated episodes of AKI has attracted increasing attention (Belayev & Palevsky, 2014; Goldstein et al., 2013). The development of experimental models that show pathophysiological characteristics similar to those seen during the transition from AKI to CKD in humans is vital to understand the progression of the disease and to generate strategies that allow an early diagnosis. Several models of the AKI–CKD transition have been described, including those using IRI or nephrotoxic substances such as cisplatin or aristolochic acid.

In the present study, we focused on evaluating FA as an inducer of kidney damage. High doses of FA have been used to induce an acute renal reaction to the generation of FA crystals that precipitate in the renal tubules (Klingler, Evan & Anderson, 1980) and produce acute tubular necrosis followed by epithelial regeneration and cortical healing (Fink, Henry & Tange, 1987). In addition, it has been shown that FA has nephrotoxic activity at various levels of the nephron because it induces a pro-oxidant state by increasing lipid binding and reducing protective anti-oxidant enzymes (Gupta et al., 2012). It has been suggested recently that the residual structural damage that is produced by FA could lead to CKD pathology (Fu et al., 2018).

Creatinine and BUN levels have been used traditionally in clinical practice as markers of renal function. Using a surgical model, Ormrod & Miller (1980) established three levels of renal damage based on BUN levels: mild (40–80 mg dL−1), moderate (100–200 mg dL−1) and severe (>200 mg dL−1). These levels were used in the present study for the first time as references to categorize kidney damage induced by FA. The damage was confirmed by blinded histopathological analysis with characteristic findings for each of the phases evaluated.

In the search for new markers of kidney damage, several studies have described the usefulness of Cys-C as a marker of renal function and KIM-1 and NGAL as markers of kidney damage (McMahon & Waikar, 2013; Shlipak & Day, 2013). KIM-1 has been shown to be highly expressed after a kidney damage event; in AKI, it plays a protective role in the modulation of the surviving tubular cells, although its sustained expression promotes fibrosis and the development of CKD (Bonventre, 2014). In a single-phase study of renal damage induced with adenine, an increase in urinary KIM-1 levels compared with the control group was reported (Nemmar et al., 2017). Studies using models of renal damage induced by IRI reported an increase in the expression of the Hvrc1 gene that codes for KIM-1 relative to the control group in all phases analyzed; however, its value decreased as time progressed (Ko et al., 2010; Le Clef et al., 2016). In the current study, the serum levels of KIM-1 were observed to increase as kidney damage progressed. Although IRI studies reported that the production of this protein stops during the progression of the disease because of atrophy of the renal tubules, the histological findings in the present study show that the degree of renal tubule damage induced by FA was less severe than that induced by IRI, which might explain why the protein continued to be produced. NGAL is another marker of kidney damage and its synthesis in renal tubular cells has been correlated with the prevention of damage by regulation of the expression of anti-oxidant enzymes (Mori et al., 2005). Models that have evaluated a single phase of CKD induced by adenine demonstrated an increase in serum NGAL levels relative to controls (Al Za’abi et al., 2018; Ali et al., 2018, 2017). Another study that evaluated a three-phase model of CKD induced by adenine reported a significant increase in serum and urinary NGAL only in the presence of the damage inducer (Gil et al., 2016). In the present study, it was also observed that the serum levels of NGAL increased significantly in the presence of FA; however, as the degree of damage advanced, the NGAL levels decreased even in the presence of the damage inducer. This is consistent with the results of studies that evaluated IRI-induced CKD, in which the expression of the Lcn2 gene that codes for NGAL decreased with increasing time post-ischaemia (Ko et al., 2010; Le Clef et al., 2016). Cys-C has been proposed as a marker of renal function because it is freely filtered by the glomerulus and subsequently reabsorbed and degraded by proximal tubular cells (McMahon & Waikar, 2013). Several studies of adenine-induced kidney damage have reported an increase in serum levels of Cys-C compared with the control group (Al Za’abi et al., 2018; Ali et al., 2015, 2018, 2017; Bokenkamp, Ciarimboli & Dieterich, 2001; Thakur et al., 2018); however, these studies reported only one phase of kidney damage. In the present study, it was found that the serum levels of Cys-C increased in each phase as the degree of renal damage increased, although histological evaluation suggested that the damage induced by FA in this study did not affect the structure of the glomerulus. However, Cys-C does not appear to be filtered, leading to increased serum levels that are very similar in the moderate and severe phases of disease.

It has also been reported that a persistent inflammatory response contributes to the perpetuation of kidney damage. The elevation of pro-inflammatory cytokines such as IL-1β, IL-6 and TNF-α is common in CKD (Furuichi, Kaneko & Wada, 2009) and has been used as a predictor of mortality in patients with kidney disease (Castillo-Rodriguez et al., 2017). Elevation in the expression of genes for these cytokines has also been reported in various models of kidney damage including IRI (Le Clef et al., 2016), adenine (Ali et al., 2018) and 5/6 nephrectomy (Agharazii et al., 2015). In the present study, an elevation of the serum values of IL-1β, IL-6 and TNF-α was observed as the degree of renal damage induced by FA progressed; this is consistent with the histological findings of an increase in inflammatory infiltrates as the degree of kidney damage increased.

In the present study, we used an NMR metabolomic approach to investigate the key urine metabolite changes that are associated with the progression of CKD from repeated episodes of AKI. PCA analysis and comparison of spectra showed that changes related to energy metabolites, amino acids and organic osmolytes are associated with the progression of the AKI–CKD transition. By combined evaluation of scores and loading plots, it was possible to identify 11 metabolites from a total of 23 that were responsible for sample clustering and therefore helped differentiate between groups. These metabolites were acetate, allantoin, citrate, creatine, creatinine, glycine, lactate, PAG, succinate, taurine and TMAO. Variations in the quantity of these metabolites within a group may also explain a particular behavior within the cluster; for example, the scattering of samples in the FAIG-S group may be due to an increase in the NMR intensity signal of acetate and creatine, associated with decreased intensity of allantoin, creatinine and taurine.

Four metabolites associated with energy metabolism, such as acetate, citrate, lactate and succinate, were found to contribute in the cluster separation of the FAIG-Mi group. Meanwhile, citrate and acetate contribute for the cluster of FAIG-Mo and FAIG-S, respectively. Previously, some metabolomics studies of renal interstitial fibrosis, nephrogenic diabetes, IRI and cisplatin nephrotoxicity, had reported an elevation of acetate and lactate in urine, coupling it with renal cell stress or injury (Hauet et al., 2000; Hwang et al., 2009; Pariyani et al., 2017; Ryu et al., 2019; Zhao et al., 2016). This is consistent with our findings. We found that citrate intensity signal decreased. Decreased urinary citrate was also observed in an experimental model of aristolochic acid, cisplatin nephrotoxicity and IRI (Jouret et al., 2016; Pariyani et al., 2017; Zhao et al., 2015). This is consistent with previous findings in patients with advanced CKD, because CKD generates metabolic acidosis that is counteracted by the reabsorption of citrate, which is used as an organic base (Posada-Ayala et al., 2014).

Amino acids are excreted in low amounts in urine. However, increased intensity of signal of two metabolites related to amino acid metabolism, creatine and glycine, separate the cluster of FAIG-Mi, while creatine alone contribute for FAIG-S. Increased urinary glycine was also observed in cisplatin nephrotoxicity rats (Pariyani et al., 2017; Ryu et al., 2019). It is possible the result is from injury in the proximal tubule, given that an increased presence of glycine in urine has been described in animals treated with proximal tubule toxins, or with pathologies affecting the proximal tubule (Zuppi et al., 1997). Moreover, another study assessed the association between urinary metabolites, genetic variants and CKD in humans, and observed that urinary glycine and histidine are associated with incident CKD (McMahon et al., 2017). In this study, a significantly increased signal of creatine was observed in FAIG-S group. Increased urinary levels of creatine has been reported in experimental models of adenine, cisplatin and, IRI (Jouret et al., 2016; Pariyani et al., 2017; Ryu et al., 2019; Zhang et al., 2015). A previous study of patients with metabolic acidosis showed that an increase in creatine excretion may be due to a defect in tubular reabsorption (Davies et al., 1990).

Organic osmolytes such as taurine and by-products of the methylamine metabolism pathway (trimethylamine, TMAO and dimethylamine (DMA)) are small solutes used by cells to maintain cell volume, and can serve as anti-oxidants (Yancey, 2005). Decreased levels of urine excretion of osmolytes have been reported in experimental studies (Zhao et al., 2016, 2011) and in patients with CKD (Posada-Ayala et al., 2014). In this study, a reduction in the levels of these osmolytes was observed as the degree of renal damage induced by FA progressed.

Phenylacetylglycine is a metabolite influenced by gut flora. Prior studies using models of adenine and aristolochic acid renal damage, have reported an increased excretion of PAG in urine (Zhang et al., 2015; Zhao et al., 2015). In this study, the signal of PAG was one of the variables contributing to the separation of the FAIG-Mo group, but its intensity in urine decreased in comparison to the control group. It is possible that changes in PAG reflected an alteration in the microbiome, as it has already been described in previous studies (Vaziri et al., 2013).

Allantoin is a product of the oxidation of uric acid or of purine metabolism. It has been described as a potential biomarker of kidney injury in experimental models, such as aristocholic acid, unilateral ureteral obstruction and cisplatin (Pariyani et al., 2017; Zhang et al., 2018; Zhao et al., 2015). The reduction in urinary allantoin excretion contributes to FAIG-S clustering. Recently, allantoin, ribonate and fumarate have been associated with high mortality rates in CKD, and the increased levels of allantoin in blood has been linked to an increase in the oxidative stress status in patients (Suzuki & Abe, 2018).

To further support these results, comparison of different models on different analytical platforms (MS and NMR), was conducted by means of a pathway analysis on the basis of the differential metabolites in the three groups hitherto described. It was found that some metabolic pathways were shared by some models, such as the citrate cycle, and taurine and hypotaurine metabolism. The same results had been previously reported in experimental models of aristocholic acid and cisplatin (Li et al., 2017; Zhang et al., 2016; Zhao et al., 2015). Simultaneously, glycine, serine and threonine metabolism, together with arginine and proline metabolism, had been reported in adenine and cisplatin model, respectively, (Li et al., 2017; Zhang et al., 2015).

Conclusions

The characterization of this model of kidney disease induced by repeated doses of FA in rats by analysis of biochemical markers, histopathology, pro-inflammatory cytokines and kidney injury biomarkers revealed that this model can be used to study the transition between AKI and CKD, and that the transition can be divided into three well-differentiated phases. The metabolomic analysis of rat urine made it possible to identify potential biomarkers for the diagnosis of the different phases of kidney disease, including acetate, creatine, creatinine, DMA, hippurate, glycine, lactate, PAG, succinate, taurine and TMAO. The results demonstrate that it was mainly metabolites involved in the energy and amino-acid pathways, together with the organic osmolytes, that were deregulated, leading to the progression of kidney disease. Future research is needed to validate these potential biomarkers.

Supplemental Information

Supplemental Information 1 Biochemical markers, pro-inflammatory cytokines and kidney injury biomarkers.

Click here for additional data file.

Additional Information and Declarations

Competing Interests

Author Contributions

Animal Ethics

Data Availability

The authors declare that they have no competing interests.

Marlene Marisol Perales-Quintana conceived and designed the experiments, performed the experiments, analyzed the data, prepared figures and/or tables, authored or reviewed drafts of the paper, approved the final draft.

Alma L. Saucedo conceived and designed the experiments, performed the experiments, analyzed the data, contributed reagents/materials/analysis tools, prepared figures and/or tables, authored or reviewed drafts of the paper, approved the final draft.

Juan Ricardo Lucio-Gutiérrez conceived and designed the experiments, performed the experiments, analyzed the data, contributed reagents/materials/analysis tools, prepared figures and/or tables, authored or reviewed drafts of the paper, approved the final draft.

Noemí Waksman contributed reagents/materials/analysis tools, authored or reviewed drafts of the paper, approved the final draft.

Gabriela Alarcon-Galvan analyzed the data, authored or reviewed drafts of the paper, approved the final draft.

Gustavo Govea-Torres performed the experiments, prepared figures and/or tables, approved the final draft.

Concepcion Sanchez-Martinez conceived and designed the experiments, performed the experiments, analyzed the data, prepared figures and/or tables, approved the final draft.

Edelmiro Pérez-Rodríguez conceived and designed the experiments, contributed reagents/materials/analysis tools, approved the final draft.

Francisco J. Guzman-de la Garza conceived and designed the experiments, contributed reagents/materials/analysis tools, approved the final draft.

Paula Cordero-Pérez conceived and designed the experiments, performed the experiments, analyzed the data, contributed reagents/materials/analysis tools, prepared figures and/or tables, authored or reviewed drafts of the paper, approved the final draft.

The following information was supplied relating to ethical approvals (i.e., approving body and any reference numbers):

All animal experiments were performed in accordance with the principles and regulations of the Mexican Official Norm NOM-062-ZOO-1999 and were approved by the Animal Care and Use Committee of Universidad Autónoma de Nuevo León (HI17-00004).

The following information was supplied regarding data availability:

The raw data is available in Dataset S1.

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
