# Peer review of "Metabolomic and biochemical characterization of a new model of the transition of acute kidney injury to chronic kidney disease induced by folic acid"

_PeerJ, doi:10.7717/peerj.7113_

## Round 0.1 · original submission · Major Revisions

Your paper was reviewed by 2 independent experts in this field and there are some revisions which will be needed including some additional experiments requested from Reviewer 1. Please also address each of the enumerated Reviewer queries in your revision.

·

Basic reporting

no comment

Experimental design

no comment

Validity of the findings

no comment

Additional comments

In the present study, Marlene Marisol Perales-Quintana et al have tried to characterize and establish folic acid induced rat model of acute kidney injury (AKI) to chronic kidney disease (CKD) transition . The experimental design is good as they have looked into various inflammatory, kidney injury markers and metabolites related to CKD in serum and urine after inducing CKD with folic acid . However, some of the minor concerns are as below.

1) The group have just performed H and E staining for histology and mentioned about inflammatory infiltrate, it would be good if they could stain for specific macrophage infiltration (CD68) in the kidney tissues for assessing inflammation.
2) The group have looked into inflammatory and kidney injury markers in systemic circulation by measuring it in serum, but it would be good to look into some of the inflammatory, oxidative stress and kidney injury gene expression levels in the kidney tissue to relate it to kidney damage.
3) The authors should proofread the paper for minor grammatical errors and sentence formations.

·

Basic reporting

See the general comments for the author.

Experimental design

See the general comments for the author.

Validity of the findings

See the general comments for the author.

Additional comments

In this work, the authors developed metabolomics approach to investigate the transition of acute kidney injury to chronic kidney disease induced by folic acid-induced renal injury model. Several suggestions are made as follows to improve the quality of the manuscript.
1. Metabolomics approaches, such as UPLC-MS and HNMR, applied to AKI or CKD have been reported in the several excellent reviews such as Nat Rev Nephrol 2017,13(5),269-284; Nat Rev Nephrol 2011,8(1),22-33; Clin Chim Acta 2013,422,59-69; Adv Clin Chem 2015,68,153-175. Please summarize the metabolomics application in renal disease to improve manuscript quality.
2. Folic acid-induced AKI-to-CKD transition should be compared and discussed with other models such as ischaemia–reperfusion injury (IRI), cisplatin induced nephrotoxicity or aristolochic acid nephrotoxicity reported by several studies such as Free Radic Biol Med. 2019,134,484-497; Sci Rep 2015,5,12936; Bioanalysis 2015,7(6),685–700; Redox Biol 2016,10,168-178; Adv Clin Chem 2014,65:69-89; Chem Biol Interact 2016,252,114-130.
3. Chemicals and Reagents should be provided in this section. Please include both the manufacturer’s name and location (including city, state, and country) for specialized equipment and reagents throughout the manuscript.
4. What method was used for identifying metabolites? The identification method of metabolites should be provided including available database.
5. What method was used for normalizing metabolites in sample?
6. Better describe the metabolomics data, how many were increased and how many decreased? This information is essential to enable the reader to determine the significance of the data presented for biomarkers.
7. There is no scale bar on the histology slides in Figure 2.

---

## Round 0.2 · accepted · Accept

All Reviewer queries have been appropriately addressed. Thank you for the opportunity to Review this interesting work.

·

Basic reporting

The authors have improved the manuscript, so I suggest the manuscript should be accepted for publication.

Experimental design

Experimental design is good.

Validity of the findings

Validity of the findings is good.

Additional comments

The authors have improved the manuscript, so I suggest the manuscript should be accepted for publication.